# Facial Emotion Recognition Deficit in Children with Moderate/Severe Obstructive Sleep Apnea

**DOI:** 10.3390/brainsci12121688

**Published:** 2022-12-08

**Authors:** Fu-Jun Zhao, Qing-Wei Chen, Yunxiao Wu, Xiaohong Xie, Zhifei Xu, Xin Ni

**Affiliations:** 1Department of Otolaryngology, Head and Neck Surgery, Beijing Children’s Hospital, National Center for Children’s Health, Capital Medical University, Beijing 100045, China; 2National Center for International Research on Green Optoelectronics, South China Normal University, Guangzhou 510006, China; 3Lab of Light and Physio-Psychological Health, School of Psychology, South China Normal University, Guangzhou 510631, China; 4Guangdong Provincial Key Laboratory of Optical Information Materials and Technology & Institute of Electronic Paper Displays, South China Academy of Advanced Optoelectronics, South China Normal University, Guangzhou 510006, China; 5Beijing Key Laboratory of Pediatric Diseases of Otolaryngology, Head and Neck Surgery, Beijing Pediatric Research Institute, Beijing Children’s Hospital, National Center for Children’s Health, Capital Medical University, Beijing 100045, China; 6Division of Respiratory Medicine, Ministry of Education Key Laboratory of Child Development and Disorders, National Clinical Research Center for Child Health and Disorders, Chongqing 400014, China; 7International Science and Technology Cooperation Base of Child Development and Critical Disorders, Chongqing Key Laboratory of Pediatrics, Children’s Hospital of Chongqing Medical University, Chongqing 400715, China; 8Department of Respiratory Medicine, Beijing Children’s Hospital, National Center for Children’s Health, Capital Medical University, Beijing 100045, China

**Keywords:** obstructive sleep apnea, sleep-disordered breathing, positive classification advantage, emotional expression recognition task, schematic face, children

## Abstract

Although previous studies have reported a facial expression classification deficit among adults with SDB, we do not know whether these findings can be generalized to children. In our study, children with sleep-disordered breathing (SDB) were divided into three groups: primary snoring (*n* = 51), mild obstructive sleep apnea (OSA) (*n* = 39), and moderate/severe OSA (*n* = 26). All participants, including 20 healthy controls, underwent an overnight polysomnography recording and the Emotional Expression Recognition Task. Psychosocial problems were evaluated using the parent-reported Strengths and Difficulties Questionnaire (SDQ). There was a borderline significant interaction between expression category and group on reaction times. Further analysis revealed that positive classification advantage (PCA) disappeared in the moderate/severe OSA group, whereas it persisted in the control, primary snoring, and mild OSA groups. Emotional symptoms were positively correlated with OAHI. In both the happy and sad conditions, RT was negatively related to age and body mass index (BMI) but was independent of the obstructive apnea-hypopnea index (OAHI), arterial oxygen (SaO_2_) and total sleep time. The accuracy of identifying a sad expression was negatively related to conduct problems. Children with moderate/severe OSA exhibited dysfunction in facial expression categorization, which could potentially affect social communication ability.

## 1. Introduction

Sleep-disordered breathing (SDB) is a common illness in childhood, with an incidence of 5.1% to 13.3% [1]. SDB is a chronic sleep disorder characterized by repetitive upper airway closures, oxygen desaturation and sleep fragmentation leading to cognitive dysfunction. In recent years, mounting studies have reported the prevalence of impaired social functioning in children with SDB [2,3], such as conflict resolution and social interaction [4].

The recognition of facial expressions is a higher-level social cognitive ability. Faces and facial expressions are of fundamental importance in social communication and behavioral regulation because they provide key signals for understanding the motivations, feelings, and intentions of others [5]. Positive facial expressions are typically detected more quickly than negative facial expressions, which refers to a positive classification advantage (PCA) [6]. This phenomenon is reported to be particularly pronounced for the recognition of happy (positive) and sad (negative) expressions [7,8,9,10]. The disappearance of this effect is considered to be suggestive of an impairment of social cognitive functioning (such as emotion recognition, emotional reactivity and emotion regulation) in certain mental disorders [11], such as schizophrenia [12], insomnia [13], learning disorder [14], attention-deficit/hyperactivity disorder (ADHD) [15] and major depressive disorder [16], which might have undesirable effects on family and peer relationships.

Deficits in facial emotional information processing have been observed in both experimental acute sleep deprivation [17,18,19,20] and chronic sleep disorders [6,10] in adults. Unlike specific deficits in neurocognitive performance in the adult SDB population, children with SDB have been found to show a generalized impact on neurocognitive functioning [21]. Inadequate sleep is associated with deficits in the processing of facial information during early adolescence, a key developmentally sensitive period in socio-emotional development [22]. Previous studies have reported a facial expression classification deficit among adults with SDB [9,23]. Thus, it is possible that this deficit also exists in the pediatric SDB population. Moreover, previous studies recruiting adult patients with OSA [9,23] have failed to consider the potential role of OSA severity (mild OSA vs. moderate/severe OSA), or the severity of SDB (primary snoring vs. mild OSA vs. moderate/severe OSA). A recent meta-analysis reported that cognitive impairment exists across SDB severity (PS, mild OSA and moderate/severe OSA) [21]. The primary objective of the present study was to directly test facial expression classification performance in children with SDB of varying severity (primary snoring, mild OSA, and moderate/severe OSA). A recent study using diffusion tensor imaging found white matter alterations in the brain in children with OSA [24]. These alternations might be related to emotion processing [25]—this suggests the existence of difficulty in recognizing emotion. Thus, we predicted that children with different severities of OSA or even children with primary snoring may have an emotional recognition deficit.

Besides the behavioral pattern of emotion recognition, some correlations were also identified in previous studies, which might be the potential targets for intervention studies to improve the ability to recognize emotion. The ability to recognize emotion followed increasing linear trends over the age range of 6–16 years for happy, fear, disgust and surprise, but not for sad and angry expressions [26]. Considering the close relationship between sleep and emotion [27], sleep characteristics might be related to the emotion recognition performance. For example, long total sleep duration was positively associated with improved ability to recognize facial expression [28], although the evidence was not totally consistent [29]. Though not conclusive, the association between BMI and emotion recognition has been repeatedly reported in the previous literature, suggesting that a high BMI might impair the ability to recognize emotion [30,31]. A recent systematic review found that facial emotion recognition problems were linked to ADHD and conduct problems [32]. Employing adults with OSA as participants, Guo et al. (2020) found that the behavioral performance of emotion recognition was closely related to AHI and not to SaO_2_ [9]. Similar studies in children with OSA are still lacking. Thus, the second objective of the current investigation was to explore the correlations of emotion recognition performance. According to previous research, we hypothesized that emotion recognition performance would be positively related to age and sleep duration, while negatively associated with BMI, conduct problems and AHI.

## 2. Materials and Methods

### 2.1. Participants

We recruited 118 children with SDB and 20 age/gender-matched healthy controls without a history of habitual snoring. Patients were recruited from either the inpatient department or outpatient department at the Center of Sleep Medicine, Beijing Children’s Hospital in Beijing, China. Snoring (≥3 nights per week) was subjectively assessed by parent reports. Children who were referred to a sleep center for sleep monitoring due to habitual snoring were consecutively enrolled. Inclusion criteria were the age range of 6–14 years and the presence of habitual snoring. Children with a known severe chronic illness such as genetic syndromes, craniofacial anomalies and neuromuscular disorders or who had undergone adenotonsillectomy were excluded. None of the patients underwent any intervening treatment. All participants went to bed before 22:00 and got up after 06:00. A valid PSG recording required at least three hours of sleep time. All recruited children completed the emotional expression recognition task and the full-night standard PSG. Participants completed the behavioral tests each afternoon. Parents (predominantly mothers) provided demographic information, and questionnaires were collected on the day of the examination.

All participants were right-handed. Children were also excluded if they suffered from dyslipidemia, diabetes mellitus, neurodevelopmental delays, hypertension or congenital heart disease, or if they used psychostimulant medications.

Children provided verbal or written assent and a parent or legal guardian provided written informed consent. Research was in compliance with the Declaration of Helsinki and was approved by the ethics committee of the Beijing Children’s Hospital.

### 2.2. Procedure

The Strengths and Difficulties Questionnaire (SDQ) is a 25-item screening tool [33], which is widely used to assess different psychosocial and behavioral problems in children. [34,35,36,37] The SDQ contains five subscales containing five items each: emotional symptoms, conduct problems, hyperactivity, peer problems and prosocial behavior. Higher scores indicate more significant degrees of behavioral and emotional problems, except for prosocial behavior, for which a higher score indicates fewer behavioral problems. In the current study, we used the Chinese parent version of the SDQ, which has been well validated in previous studies [38,39].

The emotional expression recognition task [9,13] was widely used to access the processing of emotional information in both healthy participants [40] and the participants with mental disorders [12,13,16], and thus, was employed in our study to explore emotion recognition ability and directly compare with the findings in adults with OSA [9]. In this task, the participants were instructed to recognize the expressions expressed by schematic faces, which has been established to be easily recognized by preschool-aged children [41] and effectively reflect the distinct brain activation pattern when comparing emotional with neutral schematic facial expressions [42]. In the current study, the schematic face was comprised with one circle as the face, two dots as eyes, one short vertical line as the nose and one short horizontal line as the mouth. The facial expression was manipulated by the shape of the mouth (see Figure 1). Eighteen different schematic face models were employed to avoid the low levels of processing and the boredom caused by the repetition of a single face stimulus. We obtained different face models by changing the face shape, particularly the mouths.

Participants were seated in a comfortable chair in a quiet and dimly lit room to complete the test. E-prime 2.0 was used to present the task and collect responses. Stimuli were presented at the center of a computer monitor (13-inch Apple Macbook Air laptop). When viewed at a distance of approximately 70 cm, the stimuli were displayed at a size of 6.89° × 6.30° of visual angle. Each face stimulus was presented twice, which led to 108 trials (18 face models × 3 expressions × 2 times) in one block in the formal experiment. Before the formal experiment, there were 12 practice trials to help the participants become familiar with the emotional expression recognition task. If necessary, additional practice trials were allowed to ensure that children understood the procedure.

Each trial of the task consisted of the following sequence of events (see Figure 1). Face stimuli were presented for 300 ms in a pseudo-randomized order, followed by a gray rectangle of the same size as the face stimuli. In the happy or sad condition, after the emotional expression stimulus disappeared, the gray rectangle remained on the screen until participants made a response by pressing one of the labeled keys on a standard computer keyboard. In the neutral expression condition, the gray rectangle was presented for 600 ms after the neutral expression disappeared. Participants were asked to classify the three schematic facial expressions (happy, sad or neutral) by pressing the corresponding key (happy: F; sad: J; neutral: no response) on the keyboard as quickly and accurately as possible. The assignment of response keys to different emotional expressions was balanced across participants. The participants completed two blocks with a short break between blocks.

### 2.3. Overnight Polysomnography (PSG)

All children underwent standard pediatric overnight clinical PSG monitoring (Compumedics E; Compumedics, Melbourne, Australia; or ALICE 5; Philips Respironics, Amsterdam, Netherlands) at Beijing Children’s Hospital. No consumption of coffee, tea or other stimulating substances was allowed 24 h before the study. All clinical assessments and diagnoses were made by trained and experienced clinicians before PSG. Nasal airflow, pulse oxygen saturation (SaO_2_), snoring, electroencephalography, electrooculogram, submental electromyogram, electrocardiogram, heart rate, body position and thoracic and abdominal movements were recorded. PSG data were manually scored by experienced sleep technicians according to the standard guidelines of the American Academy of Sleep Medicine (AASM) [43]. Sleep data were calculated and evaluated by experienced pediatric PSG technicians. In the current study, SDB severity was categorized in accordance with the Chinese guidelines for the diagnosis and treatment of childhood obstructive sleep apnea (2020): primary snoring (OAHI < 1), mild OSA (1 < OAHI  ≤ 5), moderate/severe OSA (OAHI > 5) [44].

### 2.4. Statistical Analysis

All analyses were performed with SPSS 25.0 software (SPSS Inc., Chicago, IL, USA). Continuous data were expressed as mean ± SD or mean ± SE. Differences in continuous clinical data and PSG data were tested using ANOVA followed Turkey’s post hoc test. Categorical data were expressed as n(%). We used the chi-square test to compare categorical data. For the behavioral performance of the emotional expression recognition task, data exceeding ± 3SD were treated as outliers and were not included in the formal analysis. Repeated-measure ANOVAs were performed separately on reaction time (RT) for correct responses and accuracy rate (ACC) with group (control vs. primary snoring vs. mild OSA vs. moderate/severe OSA) as the between-participants factor and expression (happy vs. sad) as the within-participants factor. Post hoc analysis was conducted if the interaction was significant. Correlations among clinical, PSG, and behavioral data were analyzed using Pearson’s correlation coefficients for all participants. *p* values < 0.05 were considered statistically significant. The strength of correlation according to the general guidelines [42] was also provided.

## 3. Results

### 3.1. Clinical Characteristics and PSG Characteristics

The data from two participants were excluded in the formal analysis due to the outliers (one PS patient on RT_Sad_ and one mild OSA patient on ACC_Sad_). As a result, 116 children with SDB and 20 age/gender-matched healthy controls without a history of habitual snoring were included in the formal analyses. Table 1 shows participants’ clinical characteristics and PSG characteristics. We divided participants into four groups based on OAHI scores: controls (*n* = 20), primary snoring (*n* = 51), mild OSA (*n* = 39), and moderate/severe OSA (*n* = 26). The distribution of sample size for each group was similar to previous studies [45,46,47]. Age and gender did not differ between the four groups. However, the moderate/severe OSA group showed significantly higher body mass index (BMI) scores compared with the other three groups (*p* = 0.008). However, there was no difference in BMI z-score between the groups. Higher OAHI scores (*p* < 0.001), total arousal index (ArI) scores (*p* < 0.001) and lower SpO_2_ nadir (*p* < 0.001) were observed in the moderate/severe OSA group compared with the other three groups. No significant differences were found in time spent with SpO_2_ < 90% (*p* = 0.129) between the four groups.

### 3.2. Behavioral Performance

For each condition, error trials and RT below or above two standard deviations of the individual’s average RT were excluded. Behavioral results are shown in Table 2 and Figure 2. The main effect of group was not significant (*F* (1,132) = 0.318, *p* = 0.813, *η^2^* = 0.007), indicating that RTs were similar between four groups (control: 524.45 ms, standard error [*SE*] = 44.74, PS: 567.73ms, *SE* = 28.02, mild OSA: 575.88 ms, *SE* = 32.04, MSOSA: 556.47 ms, *SE* = 39.24). A significant main effect was obtained for expression (*F* (1,132) = 35.165, *p* < 0.001, *η^2^* = 0.210), indicating that RT was significantly faster for the recognition of happy stimuli (*M* = 534.46 ms, *SE* = 17.85) compared with that for the recognition of sad stimuli (*M* = 577.81 ms, *SE* = 19.42). The group × expression interaction was marginal (*F* (3,132) = 2.12, *p* = 0.100, *η*^2^ = 0.046). To further test the first hypothesis, a post hoc analysis with Bonferroni correction was applied and revealed that, in the normal control, PS and mild OSA group, the RT to happy face stimuli was faster than that for sad faces (*p* = 0.033; *p <* 0.001; *p* = 0.001, respectively), thus, reflecting obvious PCA. However, the difference on RT between the two expressions was not significant in the MSOSA group (*p* = 0.136). The first hypothesis was, thus, partially supported.

For ACC, there was no main effect of group (*F* (1,132) < 0.001, *p*  =  0.988, *η^2^* < 0.001), and no significant interaction between groups and expression (*F* (3,132)  =  0.031, *p*  =  0.993, *η^2^* = 0.001) and no main effect of expression (*F* (3,132)  =  0.785, *p*  =  0.504, *η^2^* = 0.018).

### 3.3. Correlations Results

Emotional symptoms positively correlated with OAHI (r = 0.188 (small), *p* = 0.028). For RTs, happy and sad expressions were negatively related to age (r = −0.669 (large), *p* < 0.001 and r = −0.612 (large), *p* < 0.001) and BMI (r = −0.246 (small), *p* = 0.004 and r = −0.249 (small), *p* = 0.003). For ACC, the sad face categorization was positively correlated with conduct problems (r = −0.191 (small), *p* = 0.026). All non-significant results can be found in Appendix A. The second hypothesis was, thus, partially supported.

## 4. Discussion

To the best of our knowledge, this is the first study to use the schematic facial emotion recognition paradigm to explore whether deficits exist in face processing in children with SDB and the relationship between facial emotion recognition performance and psychosocial problems. Consistent with previous studies, PCA was observed in the healthy control group. In the RT analysis, although there were no between-group differences, we found a borderline interaction between expression and group, which was consistent with our hypothesis. Further analysis revealed that the PCA disappeared only in the moderate/severe OSA group, whereas it persisted in the PS and mild OSA groups. We found that emotional symptoms were positively correlated with OAHI. For both the happy and sad conditions, RT was negatively related to age and BMI. For the sad condition, ACC was negatively correlated with conduct problems.

With increasing concern regarding the relationship between social cognitive deficits and sleep disorder, facial emotion recognition has attracted substantial recent research attention in the field of sleep medicine [48]. Facial emotion recognition is commonly studied using emotional expression recognition tasks. Previous findings suggest that facial recognition may be impaired in children with disrupted sleep [22]. A recent behavioral study of emotion recognition in adult OSA reported that the phenomenon of PCA disappeared in the OSA group, but also that there were between-group differences in RT between an OSA group and a normal control group [9]. In the current study, when children with SDB were further divided into three groups (PS, mild OSA, and MSOSA), PCA disappeared only in the moderate/severe OSA group. Comparing behavioral data between the SDB group and the normal control group revealed a trend for worse performance in the SDB group, but the difference was not significant. Some previous studies of facial emotion recognition of OSA in adults employed different facial expressions. For example, Cronlein et al. used the Facial Expressed Emotion Labelling (FEEL) test, which included six facial expressions (sadness, happiness, anger, anxiety, surprise and disgust) [23]. The results revealed that emotion recognition was impaired only in two of the six facial expressions (happy and sad) conditions in adults with OSA. Previous studies have indicated that following poor or insufficient sleep, facial expression recognition deficits were particularly evident in the recognition of happy and sad expressions [10,13,23,49]. As in these previous studies, group differences were also found between happy and sad conditions in the current study. Previous studies reported that PCA occurs in both healthy adults and children [50,51,52], and this finding was replicated in the present study. Positive facial expressions may have an advantage in visual processing because of the importance of positive expressions in emotion regulation and social relationship formation, and because they occur at a higher frequency than negative facial expressions. Positive facial expressions can facilitate communication and trust, maximize reward, and foster alliances and collaboration [53]. It has long been suggested that the accurate identification of positive facial expressions is an evolutionary adaptation that facilitates functioning in the community. Aberrant emotional face processing is associated with social relationship dysfunction. The facial emotion recognition paradigm can be used as an objective and reliable behavioral marker in studies of emotional problems in children with SDB, but also to achieve a better understanding of the mechanisms underlying emotional and behavioral problems in this population.

We also found a positive association between emotional symptoms and OAHI, which is the gold standard for the evaluation of the severity of SDB in children. This is consistent with Horne et al. [54]. They found that the mild OSA and MSOSA group have more internalizing behavioral problems (anxious/depressed, withdrawn/depressed, somatic complaints) compared with the normal control group. More research with larger sample sizes is needed in the future.

The small but significant associations were also identified for BMI with emotion recognition performance, which replicated previous findings [30,31]. These findings suggested that interventions aim at losing weight might be beneficial to increase emotion recognition ability. The absence of association between TST and emotion recognition performance replicated previous findings [29], while contradicted most previous research [28]. Thus, this null result should be confirmed in future studies.

Previous studies reported that behavioral results in adults with OSA were positively related to the apnea-hypopnea index (AHI) and SaO_2_ [9,55]. AHI and SaO_2_ are both indicators of ventilation. We did not find any significant relationships between behavioral results and indicators of ventilation. Only two large negative correlations were found between age and RT_Happy_/RT_Sad_, which was consistent with the increasing linear trend of emotion recognition ability over the age range of 6–16 years found in previous research [26]. Children’s facial recognition ability increased with age, which was consistent with a previous study [56]. It is well known that the pathological mechanisms of adult OSA and pediatric OSA are different [57]. The pathomechanisms may differ among children with SDB at different developmental stages, and follow-up studies should examine the effects of the developmental stage. The present study highlights the importance of assessing emotional symptoms in children with moderate-to-severe SDB.

In addition to the correlation between PSG parameters and emotional recognition performance, we also aimed to explore the relationships between social problems and emotional recognition performance. The negative association between conduct problems and ACC_Sad_ found in the current investigation was not consistent with the findings of a previous study [15]. In that study, the fear recognition deficit was found to be inversely correlated with hyperactivity in adolescent boys with attention deficit and hyperactivity disorder (ADHD). These discrepancies might have been caused by differences in disorders (SDB vs. ADHD), age (children vs. adolescence) or gender (boys and girls vs. boys only).

The neural mechanisms underlying emotion recognition deficits in children with moderate/severe OSA might be related to white matter alterations in the brain (which are related to emotion processing [25]) identified in previous research [24]. However, the results only revealed brain white matter alterations in boys. Thus, this possibility requires further empirical investigation, and potential gender differences warrant further consideration.

The current study involved several limitations that should be considered. First, although an emotion recognition deficit was revealed for children with moderate/severe OSA, future studies will be needed to determine whether differences exist in emotion recognition performance between the moderate and severe OSA groups. Second, only happy and sad expression stimuli were employed in the current investigation. Future studies testing more categories of basic emotions (e.g., anger, fear and surprise) and complex emotions (e.g., shame, guilt and pride) may provide more comprehensive insights into the emotion recognition performance of children with SDB. The current investigation only used the static expressions, however, the processing of dynamic expressions might be different from that of static emotions, as previous research indicated [58]. It would be interesting to explore the recognition performance of dynamic emotions in the future. There are numerous tasks to assess emotion recognition performance, and a previous meta-analytic study found that task characteristics might exert influence on emotion recognition performance [59]. Thus, various task paradigms can be used in future studies to test the generalization of the current findings and the potential task-specific effects. It should be noted that some correlations were significant but small in the present investigation, thus, caution should be taken when making inferences from these correlation results; furthermore, studies with a large sample size and longitudinal design are needed to confirm the current findings and clarify the directionality of the associations.

## 5. Conclusions

In conclusion, our results indicate that children with moderate/severe OSA exhibit dysfunction of emotional expression categorization, which could potentially lead to a decline in social communication ability.

## Figures and Tables

**Figure 1 brainsci-12-01688-f001:**
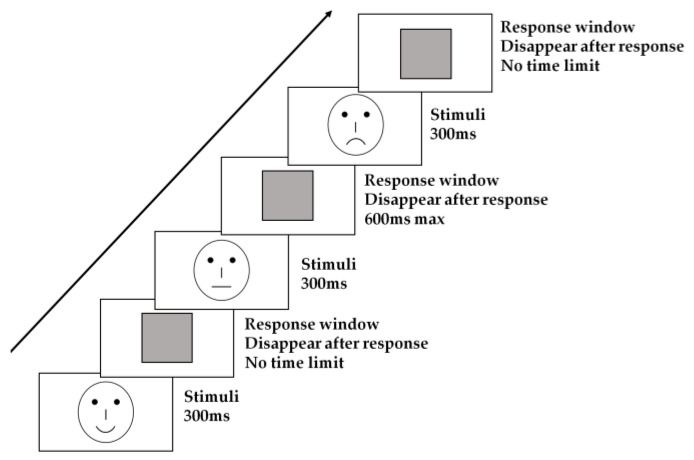
Flow chart of the experimental paradigm.

**Figure 2 brainsci-12-01688-f002:**
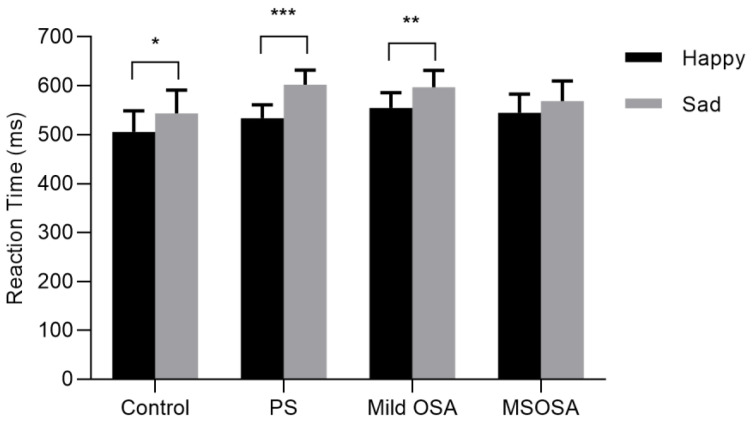
Reaction times of responses to different expression stimuli among groups. PS = primary snoring, MSOSA = moderate/severe OSA. * *p* < 0.05, ** *p* < 0.01, *** *p* < 0.001.

**Table 1 brainsci-12-01688-t001:** Clinical and polysomnographic characteristics of participants.

Characteristics	Control	PS	Mild OSA	MSOSA	*p* Value
	*n* = 20	*n* = 51	*n* = 39	*n* = 26	
Gender (%)	13M/7F	34M/17F	27M/12F	17M/9F	0.974
Age (years)	8.5 ± 1.8	8.4 ± 1.8	8.3 ± 1.7	8.6 ± 1.9	0.964
BMI (kg/m^2^)	17.5 ± 3.1 ^a^	17.5 ± 3.3 ^b^	17.8 ± 4.1 ^c^	20.8 ± 6.4	0.008
BMI z-score	0.92 ± 1.95	0.69 ± 1.19	0.85 ± 1.52	1.43 ± 1.75	0.225
OAHI (events/hour)	0.4 ± 0.3 ^a^	0.4 ± 0.2 ^b^	2.3 ± 1.0 ^c^	21.4 ± 36.1	<0.001
TST	436.7 ± 61.2	430.9 ± 57.1	419.7 ± 58.6	414.8 ± 63.2	0.508
ArI (events/hour)	5.6 ± 1.6 ^a^	5.8 ± 1.9 ^b^	6.8 ± 2.4 ^c^	16.0 ± 13.4	<0.001
SpO_2_ nadir (%)	92.8 ± 1.8 ^d^	93.2 ± 2.7 ^e^	91.6 ± 2.8 ^f^	87.4 ± 8.9	<0.001
Time with SpO_2_ < 90%	0.00 ± 0.00	0.00 ± 0.00	0.01 ± 0.02	2.08 ± 9.13	0.129
SDQ					
Emotional symptoms	2.1 ± 2.0	2.4 ± 2.2	2.6 ± 2.4	2.8 ± 1.8	0.707
Conduct problems	1.2 ± 1.0	1.5 ± 1.4	1.5 ± 1.3	1.4 ± 0.7	0.778
Hyperactivity	2.8 ± 2.2	4.1 ± 2.5	4.0 ± 1.9	4.2 ± 1.9	0.106
Peer problems	1.7 ± 1.2	2.4 ± 1.6	2.4 ± 1.7	2.4 ± 1.4	0.292
Prosocial behaviors	8.4 ± 1.4	8.1 ± 2.0	7.9 ± 1.6	8.0 ± 1.6	0.796

Values are expressed as mean ± SD. PS = primary snoring; MSOSA = moderate/severe OSA; BMI = body mass index; OAHI = obstructive apnea-hypopnea index; TST = total sleep time; ArI = arousal index. Post hoc analysis revealed significant differences between groups. *p* < 0.05 was considered significant. ^a^ = Control < MSOSA, ^b^ = PS < MSOSA, ^c^ = Mild OSA < MSOSA, ^d^ = Control > MSOSA, ^e^ = PS >MSOSA, ^f^ = Mild OSA > MSOSA.

**Table 2 brainsci-12-01688-t002:** Accuracy and reaction times of responses to different expression stimuli between the four groups (M ± SE).

	Control	PS	Mild OSA	MSOSA
RT_Happy_ (ms)	505.23 ± 43.65	533.30 ± 27.33	554.59 ± 31.26	544.70 ± 38.28
RT_Sad_ (ms)	543.66 ± 47.52	602.17 ± 29.76	597.17 ± 34.03	568.23 ± 41.68
ACC_Happy_ (%)	93.26 ± 1.14	93.19 ± 0.71	91.84 ± 0.81	93.11 ± 1.00
ACC_Sad_ (%)	93.13 ± 1.10	93.19 ± 0.65	91.95 ± 0.75	93.00 ± 0.92

PS = primary snoring, MSOS^A^ = moderate/severe OSA.

## Data Availability

The data used to support the findings of this study are available from the corresponding author upon reasonable request.

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
