# Peer review of "Facial Emotion Recognition Deficit in Children with Moderate/Severe Obstructive Sleep Apnea"

_brainsci, 2022, doi:10.3390/brainsci12121688_

Round 1

Reviewer 1 Report

Dear Editor and Reviewers,

Congratulations on your work. I would like to highlight the quality of the methodology and results section. I understood the study very well and I think it is described in a very clear and simple way. However, I found the theoretical framework very brief. I think that the authors, due to the length of the study, have summarized it as much as possible and have reviewed meta-analysis studies to give it more strength. For example, they talk about children having social difficulties but do not explain which ones, only in the recognition of emotions? Surely their ability to manage conflicts is affected. Similarly, attention may be affected by fatigue. That is why I think that a more detailed explanation would help a lot to interest readers in related areas. For example, it is a very interesting article for education and with a more detailed introduction of the ideas you have explained, it would gain a lot of quality.

In addition, I miss at the end of the introduction the objective and the hypotheses well explained. Likewise, I think it would be very interesting to add in the results when a hypothesis is confirmed or rejected. 

Reviewer 2 Report

The presented manuscript gives an important initial information on the facial emotion recognition in the younger group of patients. Unfortunately some of the data presented are not sufficiently clear. 

The first point recommended to be improved is the increased amount of abbreviations used by the authors. It would be recommended to introduce a section with description of the used abbreviations, or introduce them subsequentially. 

The second aspect of the concern is connected with the stimuli presented to the subjects. To my knowledge the stimuli were used by the authors in previous articles. It is not clear how the authors established if the stimulus is adequate in terms of emotion presented (do the stimuli originate from the standardized battery or is a self-made tool?). It would be advisable to extend the section covering this part and explain if the stimuli were morphed or present the real human faces, what degree of details are presented in the faces (just the face or significant other elements like hair). 

Thirdly, it is not clear if the authors performed the post hoc analysis on the data. The results refer to the all groups analysis. It would be advisable to add the information on post hoc analysis. As well it is not clear on what grounds authors in lines 200-206 claim the validity of the results. It would be advisable to add if the correlation ratio was mild/ moderate or not sufficiently strong to state its importance. As well it is recommended to extend the rationale of the reported relation between the RT BMI and age. What is authors view on it and why they consider it as important to report? 

It would be recommended to establish the size group effect (the groups are small and not equal in size). Please provide the additional analysis on that. 

It would be recommended to upgrade the discussion section afterwards and extend the limitation of the study section. 

Reviewer 3 Report

This study reports on findings related to acial expression recognition in children with primary snoring, mild OSA and moderate/severe OSA. Strengths of this study include the large sample size combined with PSG, in a pediatric sample. Results are undermined by apparent inconsistent reporting of conducted analyses.

Major comments

1.       Null results relating to sleep symptoms and expression processing tasks should be provided.

2.       The animal study relates to sleep deprivation but the reported effects of SDB are in sleep fragmentation.

3.       The word disease implies known pathologies.

4.       The emotional impacts of the listed mental illnesses are not listed, and these have been a rationale for the use of such tasks.

5.       The prevalence of SDB should be reported, alongside the links with other conditions and the representativeness of this sample.

6.       As a generalized impact of SDB on cognition is reported in children, there could be a greater rationale for focusing on this one task.

7.       The PSG results, especially relating to total sleep time and sleep fragmentation, should be reported and compared to age norms and guidelines, and it would have been interesting had these results been compared with the task.

8.       The ecological validity of this schematic faces should be addressed, alongside the presentation duration.

9.       The reaction times post-face presentation limit the value of these responses.

10.   The task flow in the procedure and diagram is unclear relating to which expressions were presented when.

11.   The reported age-related inclusion criterion does not match the table or figure.

12.   The interaction on the RTs was not significant, and the reported post-hoc analyses do not feature in the results section.

13.   Correlational results should be justified in the introduction, and these appear selective.

14.   Assessments appear in the discussion, but not earlier e.g. participants with ADHD, white matter results.

Round 2

Reviewer 2 Report

The authors incorporated most of the changes asked. The overall quality of the paper increased slightly. The data provided in the figure 3 are not satisfying. Please remove the data with very low correlation- it is misleading. As well the G and H plots suggest the difficulty with the raw data. Please revise the data used.
